# Effect of Lattice Constants and Precipitates on the Dimensional Stability of Rolled 2024Al during Isothermal Aging

**DOI:** 10.3390/ma16041440

**Published:** 2023-02-08

**Authors:** Rongdi Pan, Pingping Wang, Shan Jiang, Wenshu Yang, Ping Wu, Jing Qiao, Guoqin Chen, Gaohui Wu

**Affiliations:** 1School of Materials Science and Engineering, Harbin Institute of Technology, Harbin 150001, China; 2Key Laboratory of Advanced Science and Technology on High Power Microwave, Xi’an 710024, China; 3Northwest Institute of Nuclear Technology, Xi’an 710024, China

**Keywords:** rolled 2024Al, dimensional stability, lattice constants, precipitates

## Abstract

The change in material dimensional will lead to the decline of instrument accuracy and reliability. In this paper, the characterization and analysis of the lattice constant, precipitates, and dislocation density of the material by X-ray diffraction (XRD) and transmission electron microscopy (TEM) reveals the reason why the relative dimensional change in the rolled 2024Al is one order of magnitude lower than that of the as-cast 2024Al during isothermal aging. Compared with as-cast 2024Al, the dislocation density of rolled 2024Al is higher, the lattice constant decreases less before and after aging, and the precipitates have orientation and more content, resulting in the dimensional change in rolled 2024Al being smaller than that of as-cast 2024Al. In addition, two main reasons for decreasing the dimensional change in rolled 2024Al are discussed: the decrease in lattice constant, the formation and growth of the S phase before and after aging.

## 1. Introduction

With the development in aerospace, 2024 aluminum alloy has been widely used in basic components as structural materials due to its excellent mechanical properties and low density [1,2,3]. The dimensional change in this aluminum alloy structural material often requires the dimensional change to be accurate to 10^−6^~10^−7^. Beyond this range, navigation errors of aerospace equipment will occur, leading to a series of problems. Therefore, it is very important to study the dimensional change in aluminum alloy materials [4,5,6].

At present, the influence on the dimensional stability of materials is usually to observe the microstructure of materials in different treatment and heat treatment states and then connect with the macro dimensional change phenomenon to obtain the influence of microstructure on the dimensional stability [7,8,9,10,11,12,13,14]. Fu et al. studied the reasons for the reduction in the dimensional stability of high-performance extruded pure aluminum after annealing under thermal cycling. They found that the texture after annealing under thermal cycling rotated from parallel <111> to parallel <100> along the extrusion direction, and the change in grain direction led to the change in plane spacing, which further affected the dimensional stability of high performance extruded pure aluminum after annealing [15]. Dong et al. studied the influence of quenching speed on the microstructure, residual stress, and dimensional stability of the Al–Cu–Mg–Si alloy. They believed that the thermal dimensional stability of the 80 °C water-quenched samples was better than that of the 20 °C water-quenched samples due to the low residual stress of the 80 °C water-quenched sample during quenching, but the mechanical properties were not ideal [16]. To improve the dimensional stability of materials, it is very important to study the dimensional change and the micro mechanism of materials. Only by understanding the relationship between the two can we further guide the method of improving the dimensional stability of materials.

The hot rolling deformation can be regarded as the behavior in which the material achieves the ideal deformation microstructure without cracking at a given temperature and reduction [17,18]. Many researchers have made detailed research on rolled aluminum alloy and found that the texture formed during the rolling process can not only improve the mechanical properties of aluminum alloy but also improve the fatigue properties of aluminum alloy [19,20,21,22]. At the same time, Chen et al. found that the transverse and longitudinal strengths of X2095 alloy were different, and the specimen had a lower crack growth rate under longitudinal (L-T direction) loading than under transverse (T-L direction) loading [23]. There have been many studies on the mechanical properties of rolled aluminum alloys; for the application environment and purpose, it is necessary to study the dimensional change in rolled aluminum alloys, but there are very few studies in this area.

Many previous studies have shown that the precipitation of the second phase is one of the important factors affecting the dimensional stability of materials [24,25,26,27,28,29]. Song et al. believed that when an external force is applied to the Al–Cu–Mg alloy during aging, the precipitated phase S’ in 2024 aluminum alloy will be uniformly distributed with the increase in the external stress, increasing the micro yield strength of the material, thereby improving the dimensional stability of 2024 aluminum alloy, which is related to the pinning effect of S’ relative dislocation [30]. Cao et al. studied the relationship between the dimensional change behavior and the microstructure of 2024 aluminum alloy during constant temperature aging and found that the relative dimensional change during constant temperature aging was negative. They believed that one of the influencing factors was the change in lattice constant before and after aging, and the lattice strain decreased after aging. The other was the formation of the GPB region and the formation of precipitated phase S’ [4].

The purpose of this paper is to reveal the reason why the dimensional change in rolled 2024Al decreases by an order of magnitude than that of as-cast 2024Al during isothermal aging. The grain morphology of the material was observed by electron backscattering diffraction (EBSD), the dislocation density and the lattice constant were quantified by X-ray diffraction (XRD). The morphology and number density of the precipitates was observed by transmission electron microscope (TEM). The macro dimension changes were measured by using the ejector dilatometer. At the same time, the reason for the reduction in the rolling direction dimensional of rolled 2024Al is also revealed. This study provides basic data and important insights into the dimensional stability mechanism of rolled precision instrument structure materials.

## 2. Materials and Methods

### 2.1. Materials Fabrication Process and Heat Treatment

The rolled 2024Al used in this paper is purchased from Northeast Light Alloy Co., Ltd. (Harbin, China). As-cast 2024 aluminum alloy is made of rolled 2024 aluminum by semi-solid die casting.

The heat treatment process for rolled 2024Al is water quenching immediately after being dissolved in a 495 °C salt bath furnace for 1 h, and the time should be within 10 s. The samples used for real-time detection of dimensional changes during constant temperature aging shall be immediately put into the thermal dilatometer (Netzsch Co., Ltd., Selbu, Germany), and the test environment is 190 °C constant temperature aging for 72 h. The test equipment is shown in Figure 1. The sample is a cylindrical sample cut on the raw material block by electrical discharging machining (EDM), and the sample shape is Φ6 × 25 mm. The samples used for TEM observation and XRD detection shall be immediately put into a 190 °C drying oven. According to the time required for the test, the constant temperature aging time is 2 h, 4 h, 15 h, 24 h, and 72 h, respectively.

### 2.2. Testing Methods

The accurate content of the alloy is tested with the X-ray fluorescence spectrometer (XRF) of Northeast Light Alloy Co., Ltd. The diameter of the sample is 40 mm and the height is 20 mm. The upper and lower surfaces of the sample are required to be flat. Before the test, the sample is washed with alcohol. During the test, three points are taken on each sample surface to detect to reduce the error. The accurate composition of the raw material rolled 2024Al is shown in Table 1.

The microstructure and precipitates of the sample after heat treatment were observed by field emission transmission electron microscopy (TEM, JEM-2010F, JEOL, Tokyo, Japan). TEM samples are prepared by electrolytic double-spray thinning. The thinning solution is a mixture of 30% nitric acid and 70% methanol solution. During the operation, liquid nitrogen is used to cool the electrolyte. The temperature is controlled below −25 °C, the voltage is about 12~15 V, and the current is about 70 mA.

The grain morphology and texture were measured by electron backscatter diffraction (EBSD, SUPRA55), and the sample dimensional was 6 mm × 5 mm × 2 mm, polished with metallographic abrasive paper until the surface is smooth and bright without scratches. Next, the sample shall be electropolished to remove the surface stress layer and micro scratches. After polishing, the test shall be completed as soon as possible to prevent sample oxidation.

X-ray diffractometer (XRD, Empyrean, Cu K radiation, Bragg–Brentano focusing scheme) to detect phase composition and the change in lattice parameter, scanning angle of 20° to 100°, scanning speed of 0.2°/min, tube voltage 40 kV, tube current 40 mA. The residual stress is also removed by electrolytic polishing before the XRD test. Diffraction peak fitting of the sample, instrument influence, Kα1 and Kα2 peak separation, half-width, lattice constant, etc., are all realized and obtained by Jade 5.0.

## 3. Results and Discussion

### 3.1. Dimension Change during Constant Temperature Aging

The real-time curve of dimensional change in as-cast 2024Al and along the rolling direction of rolled 2024Al during 190 °C isothermal aging after solution quenching is shown in Figure 2. It can be observed from Figure 2 that the dimensional change in the rolling direction of the plate at 190 °C shows a decreasing and gradually stable trend, which can be roughly divided into three stages: the first stage of aging is 0~10 h, the dimensional change in the rolled plate shows a sharp downward trend, the second stage aging is 10~36 h, at which time the dimensional change gradually slows down, and the third stage aging is 36~72 h, at which time the dimensional change in the rolled plate is gradually stable. Finally, the relative dimensional change in the rolled 2024Al after 72 h stable temperature aging is −2.4 × 10^−4^. It can also be seen from Figure 2 that the as-cast 2024Al also shows a trend of rapid decline first and then gradually stabilizing, and the dimensional change after 72 h of aging is −7.4 × 10^−5.^

Cao et al. studied the reason for the dimensional change in isotropic 2024Al during isothermal aging and believed that the main influencing factors were the change in lattice constant and precipitates [4]. The comparison between rolled 2024Al and as-cast 2024Al shows that the main differences between the two are grain morphology, lattice constant, dislocation density, and the orientation and number of precipitates in the rolling direction. These four factors have different effects on dimensional change, which are discussed separately later.

### 3.2. The Grain Morphology of the Rolled 2024Al

The grain morphology of the raw rolled 2024Al in three directions was observed by EBSD, and the initial morphology is shown in Figure 3. Figure 3a shows the cross-section of the raw material, with good equiaxed grains. Figure 3b,c show that the grains on the parallel rolling plane and the longitudinal plane are elongated and deformed under the rolling action. There are some fine grains in the fiber shape at the same time. The length direction of the elongated grains is almost parallel to the rolling direction, which is consistent with the results observed in the literature [17,20]. During rolling, the grains will be elongated along the deformation direction to produce texture, so that certain crystal planes and crystal directions of each grain are parallel to the rolling direction (preferred orientation), as shown in Figure 3.

It is well known that the recovery process of alloys is a process in which metal atoms diffuse in a short distance to reduce dislocation density and lattice distortion, but the shape and size of deformed grains remain unchanged, and the fiber structure still exists. When the recovery process is low-temperature recovery, it mainly involves the movement of point defects [31]. The heating temperature of low-temperature recovery is 0.1~0.3 Tm of the metal melting point, and that of 2024 aluminum alloy is 66~198 °C. The constant temperature aging temperature in this paper is 190 °C, which can be regarded as the low-temperature recovery temperature. Therefore, the shape and size of deformed grains remain unchanged during aging. We believe that the shape of the texture has no significant effect on the dimensional change in the 2024 aluminum alloy sheet during isothermal aging. It cannot be considered the main reason for the difference in dimensional change between rolled 2024Al and as-cast 2024Al.

### 3.3. Measurement of Dislocation Density

In order to obtain the difference in dislocation density between as-cast 2024Al and rolled 2024Al and its influence on dimensional change, the state of the two materials after solution quenching was detected by XRD, and the dislocation density was calculated. The dislocation density was analyzed by XRD WH method [32].

The change in crystal plane spacing caused by microstrain can be expressed by the average effective microstrain *e*:(1)$$\delta_{e,hkl}=2etan\theta_{hkl}$$

*δ_e,hkl_* is the width at half maximum of the crystal plane diffraction peak caused by microstrain, *θ_hkl_* is the diffraction angle. The broadening of diffraction peaks caused by grain refinement can be expressed as follows:(2)$$\delta_{D.hkl}=\frac{\lambda }{\begin{array}{c} D_{hkl}cos\theta \end{array}}$$

*D_hkl_* is the grain size, *δ_D,hkl_* is the width at half maximum of the diffraction peak on the crystal plane caused by grain refinement, *λ* is the wavelength, and the value is 0.15418 nm which is a constant. Williamson and Hall believed that the diffraction peak broadening caused by grain refinement and the diffraction peak broadening caused by micro-stress should meet the following linear superposition relationship [33].
(3)$$\delta_{hkl}=\delta_{D,hkl}+\delta_{e,hkl}$$

Among them, *δ_hkl_* is the measured width at half the maximum of the diffraction peak on the crystal plane. Take Equations (1) and (2) into Equation (3) to obtain:(4)$$\delta_{hkl}\frac{cos\theta_{hkl}}{\lambda }=\frac{1}{D}+2e\frac{sin\theta_{hkl}}{\lambda }$$

Making various diffraction peaks *δ_hkl_cosθ_hkl_/λ* and 2*sinθ_hkl_/λ* figure, the curve slope is obtained after linear fitting, and the slope is the average effective microstrain. If the average effective microstrain is generated by dislocation, Williamson and Smallman believe that the dislocation density and the average effective microstrain are related as follows [34]:(5)$$\rho =16.1\frac{e^{2}}{b^{2}}$$
where *b* is the Bernstein vector, and for 2024Al, *b* = <110> a/2.

The rolled and as-cast 2024Al samples were subjected to XRD tests after solution quenching at 495 °C, and the results are shown in Figure 4. Read the half-width and diffraction angle of all diffraction peaks from Figure 4, and draw *δ_hkl_cosθ_hkl_/λ* and 2*sinθ_hkl_/λ*. The curve is shown in Figure 5. It can be seen from Figure 5 that the slope of rolled 2024Al is larger than that of as-cast 2024Al, which indicates that the microstrain of 2024Al in the microstrain rolling state changes greatly. This is because rolled 2024Al is deformed, so the microstrain will be larger than the as-cast 2024Al. The calculated dislocation density and microstrain are shown in Table 2. In this paper, the dislocation density, lattice constant, and other research objects are Al matrix, the content of precipitates is less and the formation is a dynamic process, so the change in its lattice constant is not explored. It can be seen that the dislocation density in the rolled plate is significantly greater than that in the as-cast 2024Al. Dislocations have many effects on materials. Those dislocations affect the precipitation of the second phase, providing preferential diffusion channels and nucleation sites for the nucleation of the second phase, and improving the formation rate of the precipitates. At the same time, the formation of precipitates will pin dislocations, making it difficult for dislocations to slip. Therefore, compared with the as-cast alloy, the rolled 2024Al has more nucleation positions of precipitates at the beginning of aging, and the precipitates are easier to nucleate, and the precipitates will form faster and generate more quantities, resulting in larger relative dimensional changes.

### 3.4. Changes in Lattice Constants before and after Aging

In order to obtain the influence of the change in lattice constant on the dimensional change in the rolled 2024Al before and after constant temperature aging along the rolling direction at 190 °C, we measured the solid solution state of the rolled plate and the XRD after aging for 72 h, and the results are shown in Figure 6. Figure 6a shows the XRD pattern of the as-rolled 2024 Al after solution and quenching. It can be seen that only the aluminum peaks can be observed in the sample at this time. In Figure 6b, it can be seen that not only the aluminum peak but also the Al_2_CuMg (S phase) peak exist in the rolled plate after aging for 72 h, which is consistent with the observation in the literature [35], indicating that Al_2_CuMg phase precipitates from the rolled plate after aging for 72 h.

Firstly, the changes in lattice constants of rolled 2024Al before and after aging were qualitatively analyzed and compared. As shown in Figure 7, the peak positions of four diffraction peaks (111), (200), (220), and (311) of aluminum aged for 0 h and 72 h were compared. It was found that the four diffraction peaks of the aged samples moved to the right. According to Bragg’s law, the lattice constant of rolled plate decreases after aging. Secondly, the changes in lattice constants before and after aging were quantitatively compared. The lattice constants of samples aged for 0 h and 72 h were accurately calculated using Jade5 XRD analysis software, and the results are shown in Table 3. The average lattice constants of the rolled 2024Al before and after aging are 0.40544 nm and 0.40392 nm respectively, and they decrease by 0.00152 nm after aging for 72 h.

According to Figure 2, it is found that the main dimensional changes in the two materials occur 24 h before aging. Therefore, by roughly comparing the lattice constant changes in the rolled 2024Al after aging with those of the literature, it is found that the lattice constant decreases much more than that of the as-cast 2024Al after aging [4]. Therefore, the difference in the lattice constant changes is also one of the main reasons for the difference in the dimensional changes in the two constant temperature aging materials and it is one of the factors that reduce the relative change in material dimensional.

### 3.5. Types of Precipitates in Rolled 2024Al

In order to accurately obtain the type of precipitates of the rolled 2024Al during the constant temperature aging at 190 °C, in addition to the XRD test, TEM observation was also conducted on the rolled 2024Al samples at 4 h and 72 h, as shown in Figure 8 and Figure 9. It can be seen from Figure 8 that when the aging time of the rolled 2024Al is 4 h, the precipitates are mainly two types of S phase, and the difference between them is that the habit planes during nucleation and growth are different [36]. In this paper, they are called S1 and S2, respectively. As shown in Figure 8b, phase S1 is mainly in the form of lath, which is consistent with that found by Zhang et al. [37]. The S2 phase is shown in Figure 8c, consistent with that observed by Yin et al. [38].

Figure 9 shows the HADDF image and main elements distribution of the rolled 2024Al after 72 h of aging. From Figure 9c,d, we can see that Cu and Mg are enriched at the same location, and further determine that S1 and S2 are Al_2_CuMg, which is consistent with the results obtained through XRD and Figure 8. In Figure 9e,f, we also found the second phase containing Cu, Fe, and Mn enrichment. From the literature, we know that this is an impurity phase formed during casting [39,40], the content of Fe and Mn phase is very low and the solid solubility in Al is very low at 495 °C [41,42], so we will not discuss this second phase in this paper.

In this section, we further determined that the main precipitates of the rolled 2024Al are the S phase, and there are two types, S1 and S2. As we know, the density of precipitates is different from that of the matrix, so the formation of precipitates will change the dimensional of materials. The density of the S phase is 3.55 g/cm^3^, the density of 2024Al is 2.78 g/cm^3^, and the specific volume between them is 0.078 cm^3^/g, so the relative dimensional change in materials after S phase precipitation will be reduced [43]. This is another reason for the dimensional reduction in the rolled 2024Al in the rolling direction.

### 3.6. Precipitation Behavior under Constant Temperature Aging

To obtain the main factors of the precipitates in the dimensional change in the rolled 2024Al during constant temperature aging, and the influence of the orientation and content of the precipitates on the dimensional change difference between the rolled 2024Al and the as-cast 2024Al during constant temperature aging precipitation, this section observed and counted the morphology change in the precipitates in the rolled 2024Al bright field phase and the length statistics of the lath-like S1 precipitates with the increase in the constant temperature time effect at 190 °C [44], as shown in Figure 10. At the same time, the difference in the orientation and the number density of precipitates between the two materials after aging for 24 h was observed and counted [9], as shown in Figure 11.

From the bright field phase diagram in Figure 10, we can observe that the formation of lath-like S1 has an obvious orientation similar to the rolling direction, and the number of S1 phases generated in the direction perpendicular to its phase is relatively small. The grains and intergranular substances (residual phases) will elongate along the deformation direction during the rolling process; therefore, the grain orientation will change, which leads to the formation of precipitates with certain orientation and a large number of dislocations. It has been reported in the literature that with the increase in aging time, the precipitates will continue to grow, and their length and width directions will increase [45]. From Figure 10, we can see that the number of S2 is relatively small, and due to the inhomogeneity of dislocations, the statistics of growth trend is difficult, and the number of generated precipitates does not exist in the statistical law, so the statistics of S2 precipitates are not carried out here. Therefore, this paper only counts the change in the length direction of the S1 phase with the increase in constant temperature aging time. It is found that with the increase in aging time, the length range of precipitates increases from 195~395 nm to 385~750 nm, and the average length changes from 289.8 nm to 563.5 nm. The increase in thickness is less obvious due to the limitation of the atomic plane interface [36]. Due to the limited sampling points for the TEM test, the variation range of the trend is slightly different from that of the test results, but the overall trend is consistent, which proves that with the increase in the precipitates and content of S phase, the dimensional change in rolled 2024Al decreases. The faster the growth rate of the S phase is, the faster the dimensional of the plate in the rolling direction decreases.

It can be seen from Figure 11a,b that the type and morphology of precipitates generated by the two are the same [46], the generation of precipitates from rolled 2024Al has obvious orientation [47], and the as-cast 2024Al has no orientation. According to the statistics of the number density of precipitates, it is found that the number density of precipitates of as-cast 2024Al is 5.0 × 10^9^ cm^−2^, while the number density of precipitates of rolled 2024Al is 9.1 × 10^9^ cm^−2^, obviously higher than as-cast 2024Al, as shown in Figure 11c. The different orientation of precipitates is mainly due to the existence of a certain habit plane between the formation of precipitates and the matrix, and the change in grain orientation will occur during rolling. The precipitates of rolled 2024Al are more formed in the direction close to the rolling direction, and the quantity is more than that of as-cast 2024Al, so the aging precipitation process leads to different dimensional changes in the two materials.

Based on the discussion above, we found that there are three main reasons for different dimensional changes in rolled 2024Al and as-cast 2024Al during isothermal aging at 190 °C, namely, different dislocation densities, different lattice constant changes, and different orientations and quantities of precipitates. The presence of dislocations will provide nucleation sites for precipitation and accelerate the diffusion of elements, leading to faster nucleation, the faster nucleation also leads to a greater rate of dimensional reduction in rolled 2024Al in the early aging period, and further leads to more precipitates in the rolled 2024Al; the lattice constant of rolled 2024Al decreases more before and after constant temperature aging; the rolled 2024Al precipitates in the rolling process are oriented and have a large number of precipitates, which leads to the difference in dimensional change between the two. For the dimensional change in the rolled 2024Al during isothermal aging, the main reason is the reduction in lattice constant before and after aging and the precipitation of the S phase, both of which lead to the reduction in its dimensional change.

## 4. Conclusions

In the current work, the reason why the relative dimensional change in rolled 2024Al is reduced by one order of magnitude than that of as-cast 2024Al during constant temperature aging has been investigated. At the same time, the reason why the dimensional of 2024Al as rolled is reduced at 190 °C during constant temperature aging has been studied. The conclusions are as follows:

1. The dimensional change in the rolled 2024Al along the rolling direction generally shows a decreasing trend in the process of 190 °C constant temperature aging. The dimensional decreases rapidly in 0~10h, slowly in 10~36 h, and gradually becomes stable in 36~72 h. The dimensional reduction after aging for 72 h is −2.4 × 10^−4^;

2. The dimensional change decrease in the rolled 2024Al during isothermal aging mainly has two factors: the decrease in lattice constant before and after aging and the formation and growth of the S phase, both of which are the main reasons for the dimensional reduction during isothermal aging in the rolling direction;

3. The dimensional change in rolled 2024Al is one order lower than that of as-cast 2024Al during isothermal aging mainly due to the different dislocation densities, the different changes in lattice constants, and the different orientations and quantities of precipitates. The dislocation density of rolled 2024Al is higher, the lattice constant changes more before and after aging, and the number of precipitates is more and they have orientation. These three main factors together cause the dimensional of rolled 2024Al to decrease more than that of as-cast 2024Al during isothermal aging.

## Figures and Tables

**Figure 1 materials-16-01440-f001:**
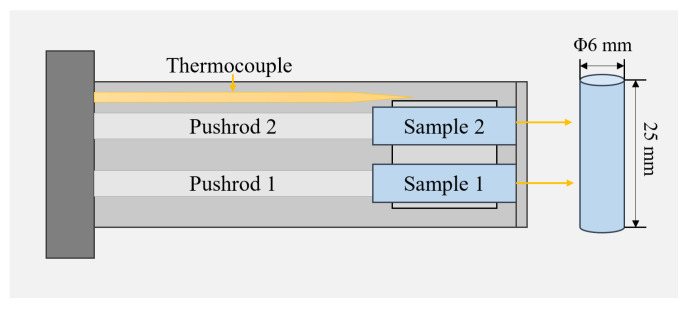
Top view of DIL402 double top bar sample chamber. The thermocouple can receive the real-time temperature in the sample chamber. The sensor connected with push rod 1 and push rod 2 can display the real-time dimensional change in sample 1 and sample 2.

**Figure 2 materials-16-01440-f002:**
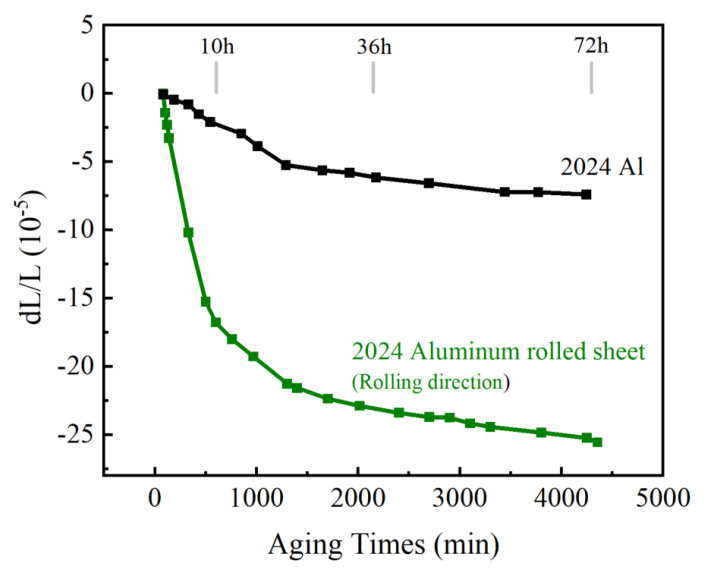
The dimensional change in rolled 2024Al in the rolling direction and as-cast 2024Al during isothermal aging at 190 °C.

**Figure 3 materials-16-01440-f003:**
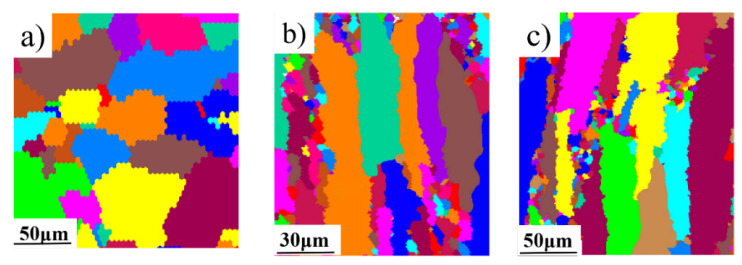
The grain morphology of the rolled 2024Al is in three directions. (**a**) Transverse section; (**b**) Longitudinal section; (**c**) Direction of rolling.

**Figure 4 materials-16-01440-f004:**
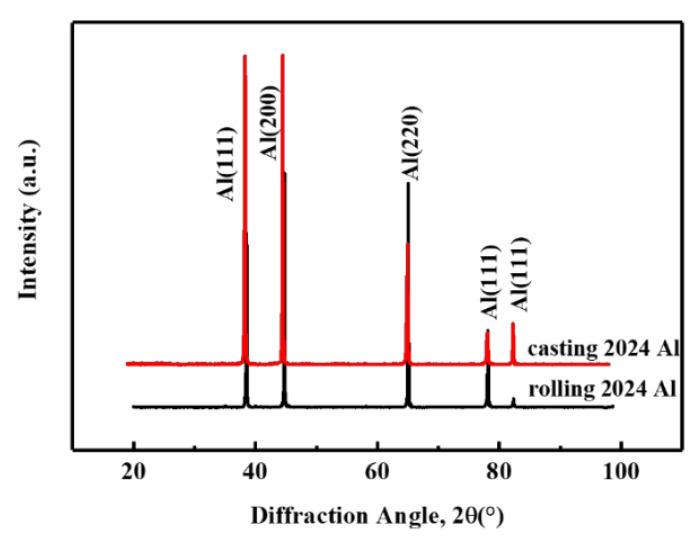
XRD patterns of as-cast 2024Al and rolled 2024Al.

**Figure 5 materials-16-01440-f005:**
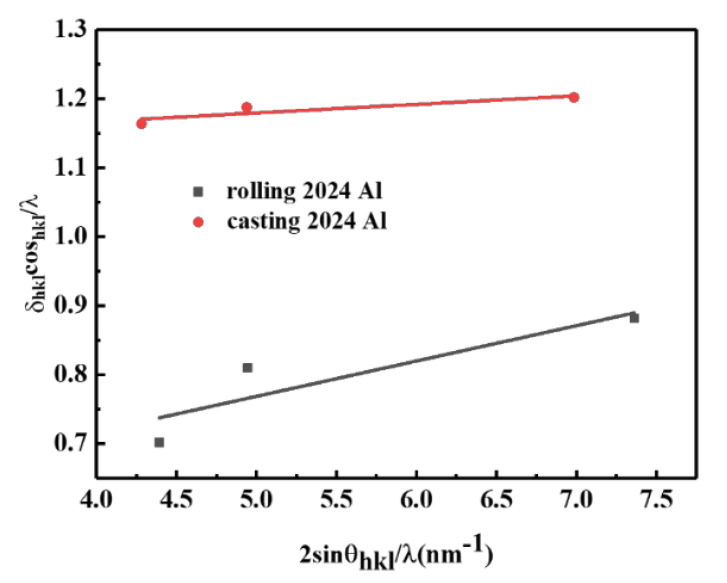
As-cast 2024Al and rolled 2024Al *δ_hkl_cosθ_hkl_/λ* and 2*sinθ_hkl_/λ* curve.

**Figure 6 materials-16-01440-f006:**
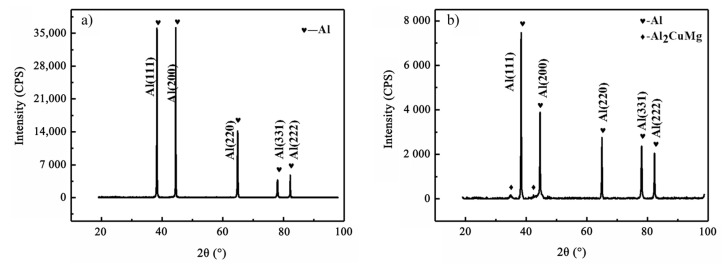
XRD test curve of as rolled 2024Al before and after aging. (**a**) Solid solution; (**b**) Aging 72 h.

**Figure 7 materials-16-01440-f007:**
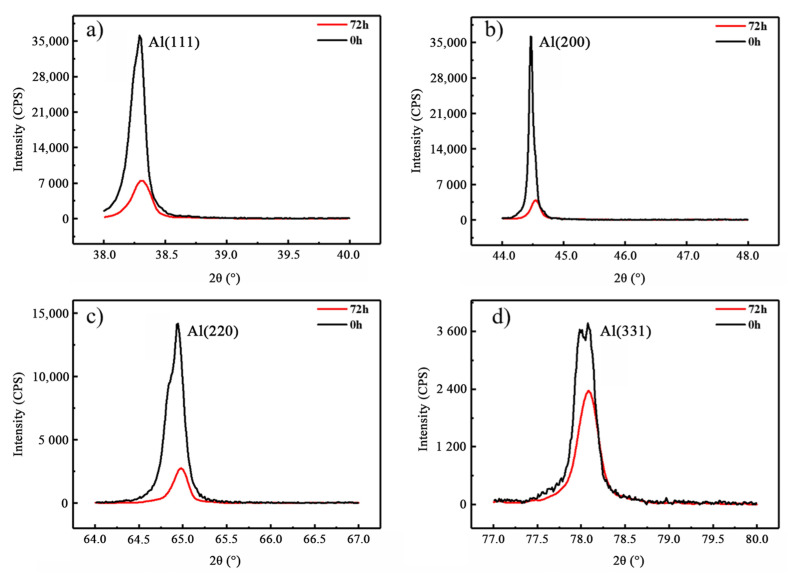
XRD comparison diagram of different Al peak positions of rolled 2024Al before and after age 72 h. (**a**) 38~40°; (**b**) 44~46°; (**c**) 64~66°; (**d**) 77~79°.

**Figure 8 materials-16-01440-f008:**
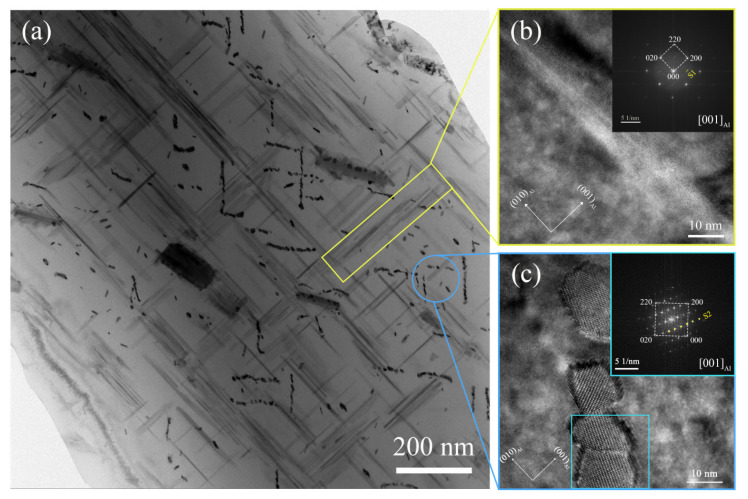
Bright-field image and high-resolution diagram of precipitates in the rolled 2024Al aged for 4 h. (**a**) Bright-field phase images of precipitates; (**b**) High-resolution image of lath-like S1 phase; (**c**) High-resolution image of the S2 phase.

**Figure 9 materials-16-01440-f009:**
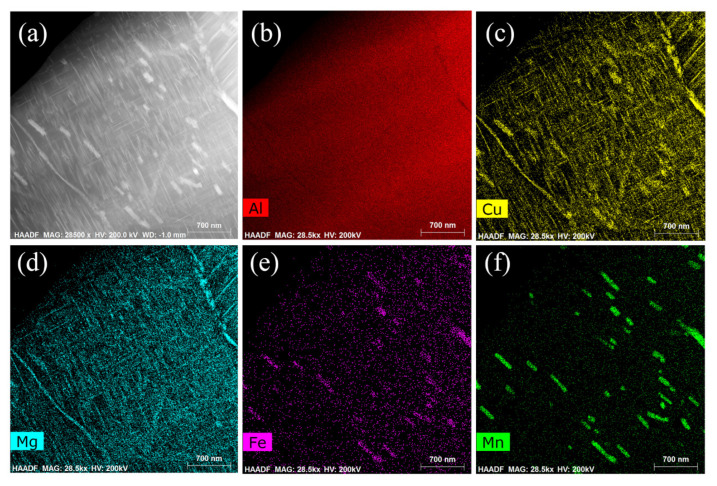
HADDF images of rolled 2024Al after aging 72 h. (**a**) HADDF images of the precipitates; (**b**–**f**) Al, Cu, Mg, Fe, Mn elements distribution map of rolled 2024Al.

**Figure 10 materials-16-01440-f010:**
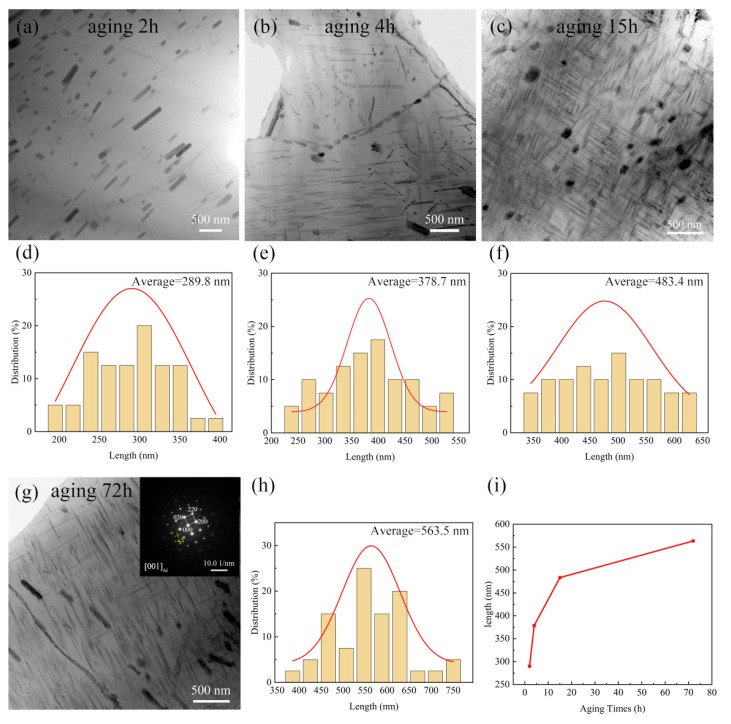
Morphology and length statistics of lath-like S1 phase at different aging times. (**a**,**d**) 2 h, (**b**,**e**) 4 h, (**c**,**f**) 15h, (**g**,**h**) 72 h, (**i**) Relationship between average length change in S1 phase and isolation aging time.

**Figure 11 materials-16-01440-f011:**
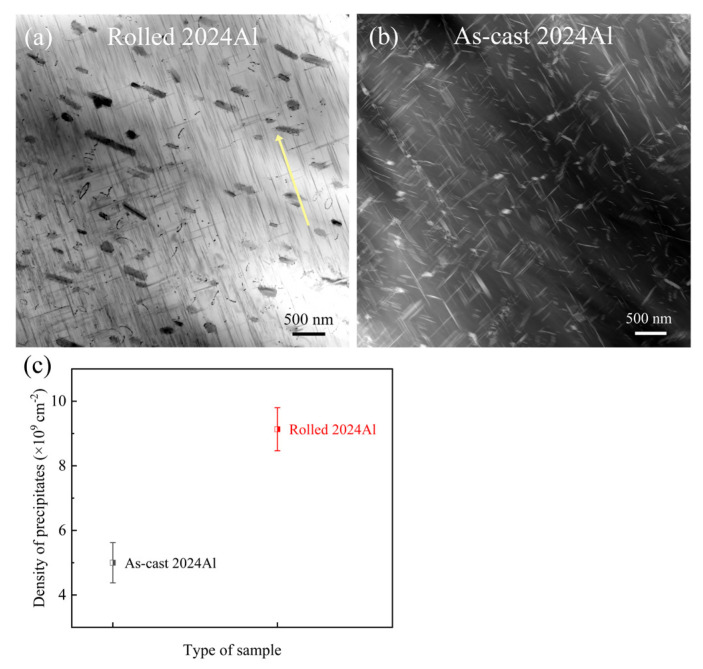
The precipitates image of rolled 2024Al and as-cast 2024Al after isothermal aging at 190 °C for 24 h. (**a**) Bright-field phase diagram of rolled 2024Al aged for 24 h, (**b**) HAADF image of as-cast 2024Al aged for 24 h, (**c**) Statistical diagram of the number density of lath-like S phase of two materials.

**Table 1 materials-16-01440-t001:** Chemical composition of 2024Al as raw material (wt.%).

Element	Cu	Mg	Si	Fe	Mn	Zn	Cr	Ti	Al
Content	4.53	1.5	0.04	0.12	0.64	0.19	0.001	0.045	Bal

**Table 2 materials-16-01440-t002:** The microstrain and dislocation density in as-cast 2024Al and rolled 2024Al after solution quenching.

Sample Type	Microstrain (*e*)	Dislocation Density (cm^−2^)
Rolled 2024Al	0.10119	9.73 × 10^12^
As-cast 2024Al	0.03242	1.45 × 10^12^

**Table 3 materials-16-01440-t003:** Change in lattice constant of rolled 2024Al before and after age 72 h.

Rolled 2024Al	Pure Al (a’)	Solid Solution State (*a_0_*)	Aging 72 h (*a*)	Δ*a*
Lattice constant (nm)	4.0494	0.40544 ± 0.00032	0.40392 ± 0.00025	−0.00152

## Data Availability

The data presented in this study are available on request from the corresponding author.

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
