# Peer review of "Effect of Lattice Constants and Precipitates on the Dimensional Stability of Rolled 2024Al during Isothermal Aging"

_materials, 2023, doi:10.3390/ma16041440_

Round 1

Reviewer 1 Report

The paper entitled «Effect of Lattice Constants and Precipitates on the Dimensional Stability of rolled 2024Al during Isothermal Aging» includes novel experimental results on structure, phase composition and dislocation density of the rolled and aged Al-bearing alloy.  The suggestions on improving the dimensional stability of 2024 aluminum alloy are given as well as the physical reasons of the dimensional changes are discussed. No doubts there are very interesting microstructure features found in this study. However, the manuscript text requires a major revision due to the following:

1) In the abstract the estimations of lattice constant by XRD and phase composition are omitted, so it’s quit difficult to reveal whether the matrix phase or precipitates exhibits changes of the lattice parameters. Also, there is no value of dislocation density. When the density of defects, lattice constant are discussed, I recommend to clarify the phase being talking about.

2) Lines 128-131, the essential information is a scheme of collecting XRD patterns. Do the authors employed a grazing incidence geometry or Bragg-Brentano focusing scheme? What formula do you use to calculate lattice parameter (cubic/hexagonal/unknown phase)?

3) Lines 182-183: «the shape of the texture has no significant effect on the dimensional change of the 2024 aluminum alloy sheet during isothermal aging». This statement should be proved experimentally or rewritten. Does it contradict with the author’s statement (lines 155-156) given before: «differences between rolled 2024Al and as-cast 2024Al are mainly the dislocation density, grain orientation, lattice constant, orientation, and the number of precipitates caused by the rolling process»?

4) I have a big request to the statement (Lines 203-204); «In general, when the grain size is less than 100 nm, the diffraction peak broadening phenomenon caused by grain refinement is relatively obvious». In my opinion, the width at half maximum of the diffraction peak corresponds to the size of coherent scattering region. I have never heard about the XRD technique to estimate the grain size. If the authors could provide a relevant citation (any book chapter or paper written by Williamson, Hall, Scherrer, Debye, etc.) I will accept your suggestion.

5) To effectively calculate the contribution to the peak broadening, it’s necessary to exclude instrumental broadening and consider the Kal-Ka2 doublet splitting. Nevertheless, these details are absent in the manuscript text. Therefore, the values listed in Table 2 are not relevant.

6) The table 3, line 273. Here, the authors provide the lattice constant of the cubic Al-matrix phase? Please, clarify and change the capture. The measurement error of the lattice constant would be a noticeable addition.

7) I may see that the peaks (111)Al, (220)Al (please, revise the notification in Fig. 7c) have a clear asymmetry, while the peak (311)Al splits into to two separate peaks. What the reason of these observations? Moreover, why are the peak intensities so different (Fig. 6)?

8) Nano-beam diffractions given in the insets (Fig. 8) are shown without indexing.

9) If the authors believe that morphology, density and size distribution of the precipitates are another reasons responsible for the dimensional reduction, consequently the volume fraction of the precipitates is also crucial. The XRD estimation of the volume fraction of the Al-phase and Al2CuMg phases observed after rolling and aging may confirm it.

Reviewer 2 Report

The paper studies the reasons for the different dimensional change of rolled and as-cast Al2024. The topic is relevant, the methodology is appropriate and the results are very interesting. However, some modifications and clarifications are needed to improve the paper quality.

General comments:

- Improve the English.

- Verify that all  symbols which appear in the manuscript, including figures and tables, are explained.

Abstract and Introduction

- In my opinion, the abstract and the introduction should be reorganized a bit. It seems that some ideas are repeated.

Materials and Methods:

- Voltage is in kV, not Kv.

Results and discussion

- Fig. 5: add comments on the scattering of the results. What can be the reason?

Round 2

Reviewer 1 Report

I would like to exprees my respect to the authors for the actual and qualified repponse to my comments. The made revisions show good work performed on improving the description and presentation. Now, the revised manuscript may be accepted for publication.